# Dual Effect of Tryptamine on Prostate Cancer Cell Growth Regulation: A Pilot Study

**DOI:** 10.3390/ijms231911087

**Published:** 2022-09-21

**Authors:** Zhuangzhuang Li, Baoyan Ding, Mustafa R. K. Ali, Lizhen Zhao, Xiaoling Zang, Zhihua Lv

**Affiliations:** 1School of Medicine and Pharmacy, Ocean University of China, Qingdao 266235, China; 2Department of Biological Engineering, Massachusetts Institute of Technology, Cambridge, MA 02139, USA; 3College of Physics, Qingdao University, Qingdao 266071, China

**Keywords:** tryptamine, prostate cancer (PCa), cell proliferation, apoptosis, cell migration, xenograft model antitumor assays

## Abstract

Abnormal tryptophan metabolism is linked to cancer and neurodegenerative diseases, and tryptophan metabolites have been reported as potential prostate cancer (PCa) biomarkers. However, little is known about the bioactivities of tryptophan metabolites on PCa cell growth. In this study, MTT and transwell assays were used to study the cytotoxicities of 13 major tryptophan metabolites on PCa and normal prostate epithelial cell lines. Ultraperformance liquid chromatography–high resolution mass spectrometry (UPLC–HRMS) was used to analyze metabolic changes in cells treated with tryptamine. Flow cytometry, confocal imaging, and Western blot were used to test the apoptosis induced by tryptamine. It was shown that tryptamine had obvious inhibitory effects on PCa cell lines PC-3 and LNCaP, stronger than those on the normal prostate cell line RWPE-1. Tryptamine was further shown to induce apoptosis and inhibit PC-3 cell migration. Metabolic changes including amino acid metabolism related to cell proliferation and metastasis were found in PC-3 cells treated with tryptamine. Furthermore, a PC-3 xenograft mouse model was used to study the effect of tryptamine in vivo. The intratumoral injection of tryptamine was demonstrated to significantly reduce the tumor growth and tumor sizes in vivo; however, intraperitoneal treatment resulted in increased tumor growth. Such dual effects in vivo advanced our understanding of the bioactivity of tryptamine in regulating prostate tumor development, in addition to its major role as a neuromodulator.

## 1. Introduction

Recently, biomarkers identified from metabolome analysis have found their additional value as bioactive compounds, e.g., exerting effects on the phenotype of the cell or organism [1]. There are several examples of endogenous metabolite-driven cell phenotype modulation; for example, intracellular arginine influences metabolic fitness and survival capacity of T cells that are crucial for antitumor responses [2], aspartate is a limiting metabolite for tumor growth and could be a target for cancer therapy [3], and butyrate suppresses cancer cell proliferation [4]. These metabolites were also reported as potential cancer biomarkers [5,6,7]. Oncometabolites that accumulate in tumor cells due to oncogenic mutations may possess oncogenic potential and affect tumorigenesis [8]. For instance, elevated levels of fumarate and succinate were reported to positively affect tumorigenesis by competitive inhibition of α-ketoglutarate-dependent dioxygenases, which consequently led to alterations of genome-wide histone and DNA methylation [9]. Elevated glycolytic flux led to an increased level of methylglyoxal, a glycolytic intermediate [8]. Addition of methylglyoxal or inhibition of its degradation enzyme glyoxalase I significantly restrained cell viability, migration and invasion and increased apoptosis in breast cancer cells by activation of the MAPK family and downregulation of Bcl-2 and MMP-9 [10]. Scheffer et al. identified mixtures of small molecules (amino acids, vitamins, nucleobases, etc.), present in serum and differentially taken up by tumor and normal cells, which exerted cancer growth inhibitory effects on Hela, MCF-7, PC-3, HepG2, Caco-2, and 4T1 cancer cell lines [8]. The components of the mixtures included tryptophan, melatonin, 5-hydroxy-L-tryptophan, arginine, etc., with concentrations ranging from 0.05 to 7.5 mM. Their findings indicated that some differential metabolites may form part of an antitumor defense system besides their role in metabolism, and altering the levels of certain metabolites in a way which is poisonous for cancer cells may be a potential therapeutic approach [8]. By identifying a cytotoxic metabolite against cancer cell lines in a metabolic pathway, its accumulation could be induced by targeting downstream enzymes and eventually poisoning the cancer cells [11]. However, only a few endogenous metabolites with cytotoxicity against cancer cells have been reported, and their effects still remain poorly understood.

Tryptophan metabolism has been regarded as a promising pharmaceutical target since its alterations have been reported in central nervous system (CNS), metabolic, infectious, and autoimmune diseases, and cancers [12]. The tryptophan metabolism pathway is shown in Figure 1. About 95% of tryptophan degradation goes through the kynurenine (Kyn) pathway [13]. In the Kyn pathway, tryptophan is converted to N-formylkynurenine (NFK) by indoleamine 2,3-dioxygenase (IDO) or tryptophan 2,3-dioxygenase (TDO). NFK is then converted to Kyn by arylformamidase (AFMID), and Kyn is oxidized, hydrolyzed or transaminated to 3-hydroxykynurenine (3-HK) or kynurenic acid (KYNA). In addition, kynureninase (KYNU) catalyzes 3-HK to 3-hydroxyanthranilic acid (3-HAA), which is converted to downstream metabolites including quinolinic acid (QA), cinnabarinic acid (CA), and picolinic acid, etc. A small fraction of tryptophan is used for protein synthesis and production of tryptamine and serotonin (5-hydroxytryptamine; 5-HT).

Neurophysiology activities of tryptophan metabolites have been intensively studied. However, little is known about the relationship between tryptophan metabolites and cancer cells, and it has aroused special interest of researchers in recent years [14]. Previous metabolomics studies have found tryptophan [15,16], Kyn [17,18,19], and 3-HAA [20] to be potential cancer biomarkers. The levels of tryptophan were reported to be significantly lower in patients with PCa and non-small cell lung cancer (NSCLC) compared with healthy controls [15,21]. A high Kyn/tryptophan ratio was correlated with the probability of NSCLC recurrence [19]. The levels of 3-HAA were found to be significantly elevated in NSCLC patients compared with healthy controls [21], while increased concentrations of 3-HAA and 3-HAA/3-HK ratios were shown to be associated with reduced pancreatic cancer risk [18]. Kyn is an oncometabolite that activates the aryl hydrocarbon receptor, a transcription factor that regulates progrowth genes, and Kyn also inhibits T-cell activity, which promotes cancer cell immune evasion [22]. The combination of 3-HAA and sorafenib was demonstrated to sensitize hepatocellular carcinoma (HCC) cells by upregulation of phosphatases PPP1R15A/DUSP6 which reduced AKT phosphorylation and activity [20]. Serotonin has a dual role in colorectal cancer and the serotonergic system can either promote or inhibit tumorigenesis through a combined modulation of immune escape and DNA repair mechanisms [23]. Serotonin could be synthesized from tryptamine, which belongs to the trace amine family, including compounds of β-phenylethylamine, tyramine, octopamine, etc., with basal concentrations ≤ 500 nM in vivo [24,25]. The discovery of trace amine-associated receptors (TAARs) in 2001 aroused much interest in the study of trace amines [26]. TAAR1 is a promising target for the treatment of mental diseases as well as a potential biomarker for cancer prognosis [27]. Tourino et al. found tryptamine and N, N-dimethyltryptamine (DMT) could increase the cytotoxic activity of peripheral blood monocytes by inhibiting IDO, while 5-HT and melatonin did not affect IDO activity [28]. A new tryptophan metabolite, 5-methoxytryptophan (5-MTP), was found to reduce COX-2-mediated cancer cell proliferation and migration by inhibition of COX-2 expression, and to reduce cancer cell invasion by inhibition of MMP-9 expression [29].

The aforementioned evidence showed that several tryptophan metabolites that were reported as potential cancer biomarkers exhibited potential anti-tumor or tumorigenesis effects, which are compound- and cell type-dependent. However, little is known about the cytotoxic effects of tryptophan metabolites on PCa cells. In our previous clinical serum metabolomics studies, we found tryptophan, indole, and indoxyl sulfate were differential metabolites for PCa detection [15,30]. Kyn, serotonin, indoxyl, and 5-hydroxy-L-tryptophan were reported to be potential PCa biomarkers [6]. Therefore, we hypothesize that metabolites of the tryptophan pathway are promising candidates for screening of endogenous anti-cancer compounds against PCa cells. In this study, we systematically studied the cytotoxicities of major tryptophan metabolites on PCa and normal prostate epithelial cell lines. Tryptamine was found to exert a significantly stronger cytotoxic effect on PCa cell lines compared to that on a normal prostate epithelial cell line, and its cytotoxic effect was also tested on other cell types. Its impact on tumor growth was further tested on a PCa xenograft mouse model. The active metabolites with cytotoxic effects on PCa cells in vitro and in vivo may advance our knowledge on the regulation of prostate tumor development and progression by endogenous compounds and possibly provide new perspectives for PCa therapy.

## 2. Results

### 2.1. Inhibition Effect of Tryptophan Metabolites on Cancer Cell Proliferation

Cell metabolomics study showed apparent distinction in metabolic profiles of PCa cell lines (PC3 and LNCaP) and RWPE-1 (Figure 2A). The tryptophan metabolism pathway was identified to be significantly altered between PC3/LNCaP and RWPE-1 cells (*p* = 0.010, Figure 2B). This corroborated our previous finding that tryptophan metabolism was altered in PCa patients compared with healthy controls [30]. We next tested the cytotoxic effects of the 13 main metabolites of the tryptophan metabolic pathway, including tryptophan, Kyn, 5-hydroxytryptophan (5-HTP), 5-HT, tryptamine, 3-HAA, melatonin, 3-HK, KYNA, QA, CA, picolinic acid, and indole, against PC-3, LNCaP, and RWPE-1 cells, and another three cancer cell lines, A549, PANC-1, and HepG2. Two other trace amine compounds, β-phenylethylamine and tyramine, were also tested for comparison. As can be seen from Figure 3A,B, most of the tested compounds reduced the proliferation of prostate and other cancer cells in a dose-dependent manner, with differences in their potency. We used etoposide as a positive control of cancer cell proliferation inhibition. The IC_50_ values of the tested compounds and positive control are summarized in Table 1.

The IC_50_ value of tryptophan on RWPE-1 (0.76 ± 0.03 mM) was slightly lower than those of PCa (1.9–6.5 mM) and the other cancer cell lines (1.5–4.5 mM), revealing that the cell proliferation inhibition effects of tryptophan were not prominent on cancer cells. For all compounds tested, tryptamine, 3-HAA, 3-HK, and CA had relatively low IC_50_ values (≤0.90 mM) for all six cell lines tested, and tryptamine exhibited significantly greater cell growth inhibition on high-metastatic PC-3 (0.24 mM) and low-metastatic LNCaP (0.037 mM) cells, compared to RWPE-1 (0.71 mM) (Table 1 and Figure 3A). These two PCa cell lines were also more sensitive to tryptamine compared with the other three cancer cell lines A549, HepG2, and PANC-1, which had IC_50_ values of 0.35–0.46 mM (Table 1, Figure 3A,B). Furthermore, as a trace amine, tryptamine exhibited a greater cytotoxicity effect compared with the other two trace amine compounds, tyramine and β-phenylethylamine, for all cell lines studied (Table 1). The anti-proliferative effect of tryptamine was also verified using CCK8 assay. The IC_50_ values of the positive control etoposide were about 1–10 times lower compared to tryptamine for all five cancer cell lines studied, and 44 times lower for RWPE-1 cells.

Interestingly, 3-HAA promoted the proliferation of RWPE-1 cells at concentrations between 0.03 and 0.1 mM (Figure 3A), while no increase in cell proliferation was observed for any other compound or cell line in this study. Kyn, 5-HT, and melatonin showed low cytotoxicity effects with IC_50_ values greater than or around 1 mM for all cell lines tested (Table 1, Figure 3A,B). Additionally, QA and KYNA were found to have little or no inhibitory effects on LNCaP and PC-3 cell growth, with IC_50_ values above 5 mM (Table 1).

### 2.2. Combined Treatment Effects of Tryptophan Metabolites on PCa Cells

The combined treatment effects of binary combinations among selected tryptophan metabolites tryptamine, 3-HAA, 5-HT, and the positive control etoposide were also evaluated on PC-3 and LNCaP cells and compared with A549, HepG2, and PANC-1 cells. For two compounds with synergic effect, the combined treatment was more effective in cell proliferation inhibition compared with either treatment alone. The combination index (CI) values are summarized in Table 2, in which 0.3–0.7 indicates synergism, 0.7–0.85 moderate synergism, 0.85–0.9 slight synergism, 0.9–1.1 nearly additive, 1.1–1.45 slight to moderate antagonism, and 1.45–3.3 antagonism [31]. The results show that the combination of tryptamine + 5-HT or tryptamine + 3-HAA had almost additive effects on the inhibition of PCa cell growth, with the CI values of 0.91 (PC-3) and 1.06 (LNCaP) for tryptamine + 5-HT, and 1.04 (PC-3) and 0.92 (LNCaP) for tryptamine + 3-HAA. However, the combination of tryptamine + 3-HAA showed synergistic inhibitory effects on the cell growth of HepG2 and PANC-1 cells, with CI values of 0.47 and 0.74, respectively. Apparent synergistic inhibitory effects on PCa cell proliferation were obtained for the combined treatments involving etoposide. The etoposide + tryptamine combination showed moderate synergism on PC-3 and LNCaP cells, with CI values of 0.78 for both cell lines, while the etoposide + 5-HT combination also showed synergism with CI values of 0.70 (PC-3) and 0.55 (LNCaP), and the etoposide + 3-HAA combination showed moderate synergism on the PC-3 cells (0.80). We should note that all these combined treatments exerted antagonistic effects on normal prostate epithelial cell line RWPE-1 to various degrees, with the CI values ranging from 1.25 to 3.03. Especially, the etoposide + 3-HAA combined treatment had a marked antagonistic effect (CI = 3.03) on RWPE-1 cell proliferation (Table 2). Furthermore, this binary combination also exhibited an apparent synergistic effect on inhibiting the growth of HepG2 and PANC-1 cells, with CI values of 0.82 and 0.51, respectively (Table 2).

### 2.3. Tryptamine Inhibited PC-3 Cell Migration

To investigate if tryptamine has any effect on PCa cell migration, we performed a transwell assay on the high-metastatic PC-3 cells, and compared it with A549, PANC-1, and HepG2 cells. The LNCaP cell line was not studied since it had low-metastatic potential. After incubation with 0.075 mM tryptamine for 24 h, the migrated cells were reduced by 22% and 43% for PC-3 and PANC-1, respectively (Figure 4A–C), while there was no significant difference in the migrated cells for A549 and HepG2 compared with the control (Figure 4D,E). Significant reduction in migration was observed for all four tested cancer cell lines when tryptamine concentration increased to 0.15 and 0.3 mM. As shown in Figure 4B–E, when tryptamine concentration reached 0.15 mM, the percentage of migrated cells was reduced to 58% (PC-3), 45% (PANC-1), 80% (A549), and 77% (HepG2), and when its concentration reached 0.3 mM, the migrated cells percentage was reduced to 23% (PC-3), 27% (PANC-1), 63% (A549), and 55% (HepG2). Additionally, no significant change in cell viability was observed after incubation of PC-3 cells with 0.15 mM tryptamine for 24 h, compared to the control group (Appendix A). Although there was a significant reduction in cell viability after treatment of 0.3 mM tryptamine for 24 h, the percentage of viable cells (75%) remained much higher than that of the migrated cells (23%), indicating that cell viability reduction induced by tryptamine had little influence on the migration reduction result for PC-3 cells. Collectively, these results showed that tryptamine could effectively reduce cancer cell migration, particularly for PC-3 and PANC-1 cells.

### 2.4. Tryptamine Induced Apoptosis in PC-3 Cells via Caspase-3-Dependent Pathway

The flow cytometry experiment showed that tryptamine induced the apoptosis of the PC-3 cells in a dose-dependent manner, with the apoptotic cell populations (early and late apoptosis) increased to 12.1% and 21.4% after 24 h incubation with 0.2 and 0.3 mM tryptamine, respectively (Figure 5A). The typical apoptotic morphological changes were observed in PC-3 cells treated with tryptamine, such as cell shrinkage and cell membrane blebbing (Figure 5B). Immunofluorescent staining (Figure 6A) and Western blot (Figure 6B) experiments showed that 0.3 mM tryptamine treatment resulted in an increased expression of cleaved caspase-3, suggesting that tryptamine induced PC-3 apoptosis via the caspase-3-dependent pathway.

### 2.5. Tryptamine Affected Tumor Growth of Mouse PC-3 Xenografts

Based on the results of in vitro study, we further investigated the effects of tryptamine on PC-3 cell growth in vivo. For the intraperitoneal treatment group, since the median lethal dose (LD_50_) of tryptamine in mice was ~200 mg/kg, we chose 80 mg/kg as an experimental dose. As shown in Figure 7A–C, the treatment group on average had increased tumor volume and weight compared with the control group.

For the intratumoral treatment group, intratumoral administration of tryptamine (1 mg/100 mm^3^) caused significant tumor growth delay with significant reduced tumor volume and weight, and even a negative growth rate of PC-3 xenograft tumors during the later treatment period (Figure 7D–F), without obvious signs of adverse effects or toxicity in the liver (Appendix A). The result was also verified in a separate experiment (Appendix A). No difference in tumor size was observed for the group with intratumoral or intraperitoneal administration of 3-HAA (Appendix A).

**Figure 5 ijms-23-11087-f005:**
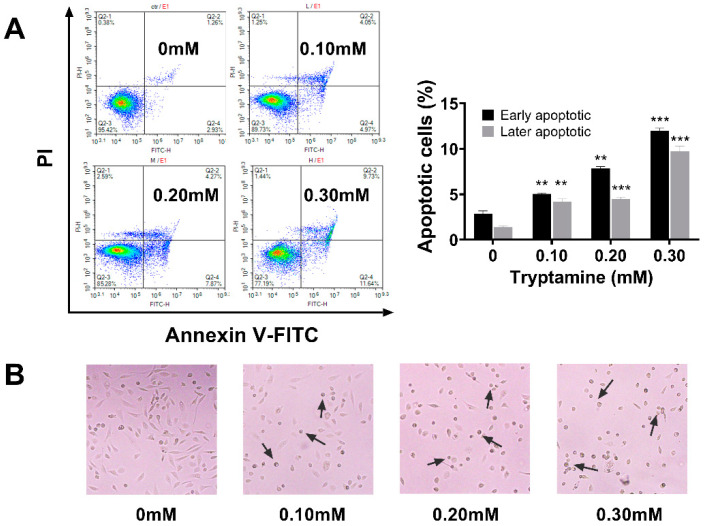
Tryptamine-induced PC-3 cell apoptosis. (**A**) Apoptosis of PC-3 cells was examined by Annexin V-FITC/PI staining and analyzed by flow cytometry after incubating with tryptamine under various concentrations (0.1, 0.2, and 0.3 mM) for 24 h. (**B**) Cell morphology of PC-3 cells after incubating with various concentrations of tryptamine for 24 h under phase-contrast microscope. ** *p* < 0.01, *** *p* < 0.001.

**Figure 6 ijms-23-11087-f006:**
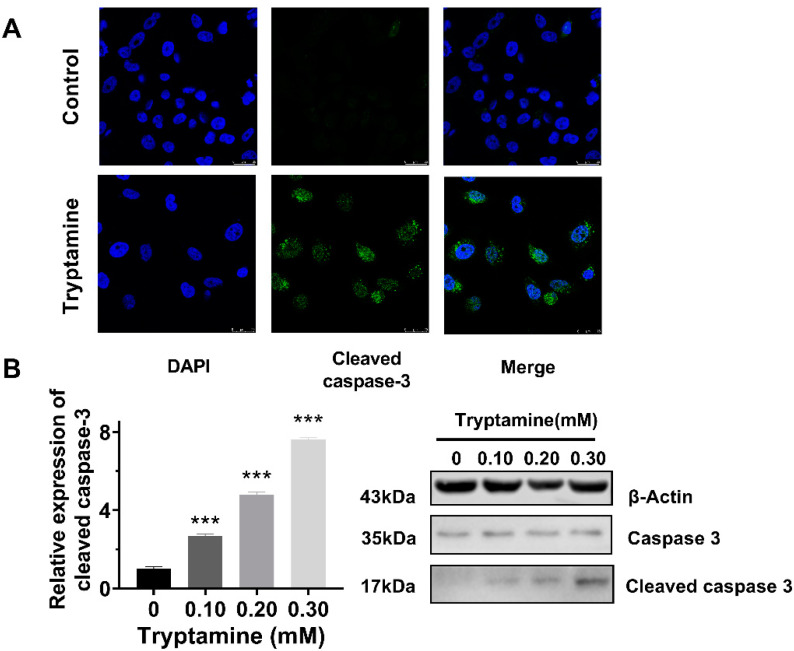
Tryptamine-induced PC-3 cell apoptosis via caspase-3 pathway verified by confocal imaging and Western blot. (**A**) Confocal imaging of cleaved caspase-3 (green) and DAPI (blue) after incubation with 0.3 mM tryptamine for 8 h. (**B**) Expression of cleaved caspase-3 protein in PC-3 cells after incubation with various concentrations of tryptamine assessed by Western blot assay. *** *p* < 0.001.

### 2.6. Tryptamine Induced Metabolic Alterations in PC-3 Cells

We performed UPLC–HRMS based metabolic profiling of cell and serum samples to study the metabolic alterations of PC-3 cells and PC-3 xenograft mice treated with tryptamine, respectively. For cell metabolomics, the heatmap showed distinct differences in metabolic profiles between the cells treated with tryptamine and the control group (Appendix A), and 11 differential metabolites (*p* < 0.05 and VIP > 1 from oPLS-DA analysis) were identified, including creatine, spermidine, spermine, 5-HT, aspartic acid, arginine, and proline, etc. (Table 3). For serum metabolomics, there was no obvious metabolic pathway alteration between the intraperitoneal/intratumoral treatment group and the control group.

**Figure 7 ijms-23-11087-f007:**
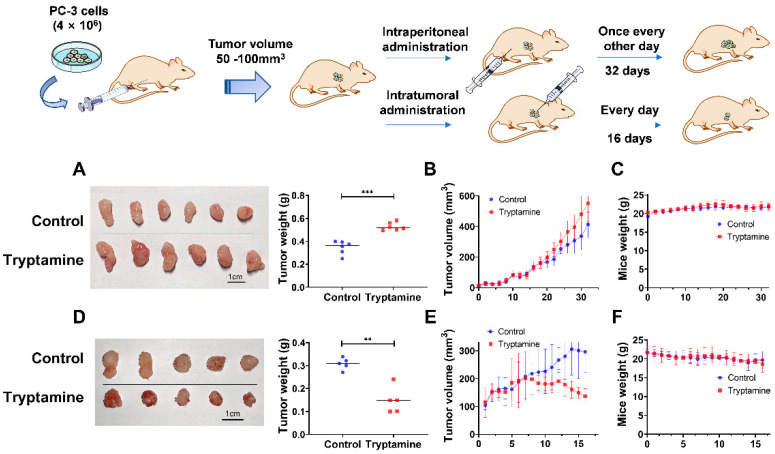
Effects of tryptamine treatments on the tumor growth of PC-3 xenograft mouse model. (**A**,**D**) Size and weight of the tumors, (**B**,**E**) growth curves of tumors, and (**C**,**F**) body weights of PC-3 xenograft mice that received intraperitoneal administration of tryptamine (80 mg/kg, n = 6) and intratumoral administration of tryptamine (1 mg/100 mm^3^, n = 5), respectively. Data are represented as mean ± SD, ** *p* < 0.01, *** *p* < 0.001.

## 3. Discussion

Besides its broad acknowledged role in biomarker discovery, metabolomics could also contribute to the identification of small-molecule metabolites capable of altering the phenotype of a cell or an organism [1]. Metabolites belonging to the tryptophan metabolic pathway are promising candidates for bioactivity screening against PCa cell growth since metabolomics studies have reported tryptophan and its metabolites as potential PCa biomarkers [15,30], and tryptophan metabolism also plays a crucial role in neurodegenerative disease, autoimmune disease, cancer, etc. [12]. There have been studies on the anti-tumor effects of the tryptophan metabolite 3-HAA [32,33,34,35], however, the effects of major tryptophan metabolites on PCa cell proliferation and tumor growth remained largely unknown. 

In this study, a total of 13 major metabolites from the tryptophan metabolic pathway and an additional two trace amines were tested to search for bioactive compounds that could impact PCa cell proliferation, and the results were compared with another three cancer cell lines (A549, HepG2, and PANC-1). Several tryptophan metabolic compounds were found to exhibit different levels of cytotoxicity with IC_50_ values ranging from ~0.04 to 1 mM, including tryptamine, 3-HK, CA, 3-HAA, and 5-HT, etc. Among these compounds, tryptamine exhibited significant stronger cytotoxicity on PC-3 (IC_50_ = 0.24 ± 0.01 mM) and LNCaP (IC_50_ = 0.037 ± 0.001 mM) compared to that in normal prostate RWPE-1 cells (IC_50_ = 0.71 ± 0.06 mM). Tryptamine is the most important compound of the indoleamine family, and its scaffold has been applied in the design of pharmacological agents with anti-migraine, antibacterial, and antitumor effects [36]. In view of the toxic risk of high concentration of biogenic amines that may accumulate in foods, Del Rio et al. used the human colon cancer cell line HT29 as an in vitro model of intestinal epithelium to study cytotoxicities of tryptamine and β-phenylethylamine, and the IC_50_ of tryptamine was 0.67 ± 0.09 mM [37], which is similar to the IC_50_ of tryptamine on RWPE-1 cells (0.71 ± 0.06 mM) determined in this study. Furthermore, tryptamine was found to induce apoptosis via caspase-3 activation in PC-3 cells in this work, and it was also demonstrated to suppress the migration of PC-3, A549, PANC-1, and HepG2 cells, suggesting its potential in inhibiting cancer metastasis. The tryptamine receptor TAAR1 is a potential cancer prognosis biomarker [27] and it could induce the transcription of the aryl hydrocarbon receptor (AHR) target gene cytochrome P450 1A1 (CYP1A1), leading to the activation of the conversion of organic compounds to cytotoxic or carcinogenic species [38]. Other metabolites had little or scarce cytotoxicity in all cancer cell lines tested, including Kyn, 5-HT, and melatonin, with IC_50_ values greater than or around 1 mM, and QA and KYNA with IC_50_ values above 3 mM (Table 1). QA is a neurotoxin, pro-inflammatory, and pro-oxidative molecule, which can stimulate the N-methyl-D-aspartate (NMDA) receptor to induce excitotoxicity, while KYNA is a powerful antagonist of QA [39], and melatonin is associated with sleep and circadian rhythm regulation in diurnal species [40]. KYNA was found to have scarce or no effect on human castration-resistant PCa cells DU145 and PC3 [41], which is in agreement with our finding of the cytotoxicity of KYNA.

Cell metabolomics revealed several compounds with significant changes after tryptamine treatment of PC-3 cells, including decreased creatine, arginine, proline, aspartic acid, spermidine, spermine, and increased 5-HT. Creatine is associated with enhanced cellular basal respiration in vitro and increased tumor cell proliferation in vivo [42]. Genetic silencing of the creatine transporter gene depletes intracellular creatine levels and decreases the colony-forming capacity of PCa cells, and in vitro treatment of PCa cells with a creatine analog cyclocreatine led to a marked decrease in intracellular levels of creatine, phosphocreatine, and creatinine and suppressed cell proliferation [42]. We found a significant decrease in the creatine level after tryptamine treatment of PC-3 cells, indicating tryptamine-induced PCa proliferation inhibition, possibly via reducing the intracellular creatine level. Arginine plays a crucial role in maintaining malignant phenotypes of multiple types of cancer including PCa, and it was shown to be necessary for the growth of PCa. Deprivation of arginine by recombinant human arginase caused significant cytotoxicity in PCa cell lines PC-3, LNCaP, and DU145 [43,44], and arginine deiminase has been studied for PCa treatment since androgen-independent PC-3 and CWR22Rv1 cells were shown to be responsive to arginine deiminase treatment by starving the cells of arginine and inducing autophagy [43,45]. Our findings of decreased arginine and proline levels in tryptamine-treated PC-3 cells is in general agreement with the reported findings, suggesting a decreased arginine level is associated with PCa cell cytotoxicity. Spermidine and spermine were also found to decrease in the treatment group in this study. Polyamines are involved in growth and proliferation of prostatic glandular epithelial cells [46]. An elevated urine spermidine level was found in PCa compared with noncancerous controls [46]. Spermine was shown to inhibit the growth of PCa cells with low-metastatic potential but not those with high-metastatic potential, indicating its regulation effects depend on the metastatic state of PCa cells [47]. The exact link between the spermine concentration and PCa cell proliferation and metastasis needs further investigation. The proline level was found to have a twofold decrease in tryptamine treated PC-3 cells compared to control (Table 3). Pyrroline-5-carboxylate reductase 1 (PYCR1), which catalyzes the last step in proline biosynthesis, was reported to be overexpressed in multiple cancers including PCa, and it had a critical function in promoting tumor progression and cancer cell survival in PCa cells, indicating the essential role of proline in cancer cell growth and survival [48]. Aspartic acid was found to have a 3.2-fold decrease in the tryptamine treated group compared to the control group (Table 3). Aspartic acid is an important metabolite of cancer cell energy metabolism [7]. Sullivan et al. reported aspartate acquisition is an endogenous metabolic limitation for tumor growth and metformin inhibited tumor growth by depleting aspartate, suggesting therapeutic aspartate suppression might be an effective cancer therapy [49]. Aspartate was reported to be positively associated with PCa development [50]. These findings corroborated our findings showing a reduced aspartate level is linked to PCa cell proliferation suppression induced by tryptamine treatment. To our surprise, the 5-HT level had a prominent spike after tryptamine treatment, while it was barely detected in untreated cells (fold change of 1.4 × 10^3^ and p value of 1.4 × 10^−3^, see Table 3). Serotonin was reported to have a stimulatory effect on PCa cell growth mediated through 5-HT_1A_ and 5-HT_1B_ receptors; however, another study did not support the hypothesis of a prominent role of 5-HT_1A_ and 5-HT_1B_ receptors in the control of PCa cell growth at physiological concentration of 5-HT [51,52,53]. The discrepancy might be due to the involvement of other 5-HT receptor subtypes that are expressed in PCa cells at high 5-HT concentration [53]. The exact mechanism of increased 5-HT in the tryptamine treatment group needs to be further investigated, and it might be simply due to the conversion of added tryptamine by tryptamine 5-hydroxylase (T5H) to 5-HT.

In the combinational treatment study, apparent synergic inhibition effects on PCa cells were found for combined treatments including etoposide. Etoposide + tryptamine and etoposide + 5-HT exhibited a synergistic inhibitory effect on PC-3 and LNCaP cells, and etoposide + 3-HAA combination showed synergism on PC-3 cells. All three combined treatments exerted antagonistic effects on normal prostate RWPE-1 cells. The joint effects of etoposide + 3-HAA, etoposide + 5-HT, and etoposide + tryptamine could be potentially utilized to reduce the cytotoxicity of etoposide on normal prostate epithelial cells, meanwhile increasing the cytotoxicity on PCa cells [20], and the results could provide insights into the clinical practice of etoposide in PCa treatment.

In vivo experiments showed that intratumoral administration of tryptamine to PC-3 xenograft tumors in nude mice caused tumor growth delay and finally led to a negative growth rate compared to control group, suggesting its potential in local anti-tumor therapy. However, intraperitoneal injection of tryptamine promoted the growth of PC-3 xenograft tumors in nude mice. One possible reason might be that locally injected tryptamine was mainly distributed inside tumors, while the administration of tryptamine treated intraperitoneally was metabolized rapidly to other forms and exerted a promotion effect on the growth of PC-3 xenograft tumors through indirect ways, since in a UPLC–HRMS experiment, tryptamine was not detected in serum or tissue samples in the intraperitoneal treatment group, while it had an apparent peak in tissue samples in the intratumoral treatment group. Additionally, it has been reported that the intraperitoneal injections of 10 mg/kg 5-HT increased tumor weight by almost 60% using an orthotopic mouse model of ovarian cancer [54]. As is known, both 5-HT and tryptamine in the blood can be taken up and carried by platelets and released during degranulation [55,56], and also various tryptamine-related compounds are substrates of the platelet 5-HT transporter [57]. Thus, the tumor growth promoting effect of intraperitoneal treatment of tryptamine in our study might be similar to the effect of 5-HT, which has long been thought to be related to tumor progression [58]. Serum metabolomics showed no obvious metabolic pathway alteration between intraperitoneal/intratumoral treatment and control groups, suggesting that the tumor volume and weight change induced by treatment of tryptamine in vivo was not associated with metabolic profile changes. 

Besides being a metabolite of tryptophan, tryptamine was also found in a variety of foods including onion-family vegetables, green zucchinis, and fruits such as acerola and custard apples [59]. In contrast to current clinical treatment choices of PCa such as surgery, hormone therapy, and radiotherapy, which might be associated with side-effects or limited in prevention of metastasis, diet and nutrient supplementation are safe with little or no side effects and are good candidates for adjunctive therapeutic treatment of PCa [60]. Previous study found the combination of ascorbic acid, lysine, proline, arginine, and epigallocatechin gallate inhibited proliferation and invasion of PCa cell lines PC-3, LNCaP, and DU145 [61], and the formulation of lysine, proline, ascorbic acid, and green tea extract supplemented to regular diet could significantly inhibit tumor growth in PC-3 xenograft mice, suggesting its potential as an excellent candidate for adjunctive therapeutic treatment of highly metastatic PCa [60]. Additionally, targeting the synthesizing enzyme of a building block metabolite to disrupt its production might not be effective in killing cancer cells when there are alternative pathways that produce the essential metabolite; however, identification of a toxic metabolite in that pathway could allow us to target the enzyme directly downstream of it, resulting in the accumulation of the upstream toxic metabolite, which could poison and kill cancer cells even when alternative pathways that can produce the building block metabolite exist [11]. According to our findings, tryptamine itself or in combination with etoposide could inhibit PCa cell proliferation and migration in vitro. Tryptamine could modulate the tumor growth of PC-3 xenograft mice, with intratumoral administration resulting in a significant reduction in tumor growth and even in tumor sizes and intraperitoneal administration resulting in increased tumor growth, without adverse effects in nude mice. Together, the results suggest the potential use of tryptamine in adjunctive therapeutic treatment of highly metastatic PCa, once validated with clinical testing.

One limitation of the study is that the mechanism of the dual effect of tryptamine in vivo is not clear, and future work on elucidating the exact molecular mechanism of the function of tryptamine in vivo is needed.

## 4. Materials and Methods

### 4.1. Chemicals

Tryptophan, 3-HK, Kyn, CA, 5-HTP, QA, picolinic acid, tyramine, indole, and etoposide were purchased from Aladdin (Shanghai, China). Tryptamine, 3-HAA, melatonin, 5-HT, KYNA, β-phenylethylamine, and tyramine were purchased from MACKLIN (Shanghai, China). MTT (3-(4,5-dimethylthiazol-2)-2,5-diphenyltetrazolium bromide) reagent was purchased from Aladdin (Shanghai, China). Other reagents involved were of analytical grade.

### 4.2. Cell Culture

PC-3, LNCaP, RWPE-1, HepG2, A549, and PANC-1 cells were purchased from the National Collection of Authenticated Cell Cultures (Shanghai, China). PC-3 and A549 cells were cultured in F-12K Nutrient Mixture (Gibco, New York, NY, USA), LNCaP in RPMI 1640 (Gibco, New York, NY, USA), PANC-1 in Dulbecco’s Modified Eagle Medium (DMEM, Gibco, New York, NY, USA), and HepG2 in minimal essential medium (MEM, Gibco, New York, NY, USA). All the above cell cultures were supplied with 10% fetal bovine serum and 1% penicillin-streptomycin. RWPE-1 cells were cultured in keratinocyte serum-free medium (K-SFM, Gibco, New York, NY, USA) supplemented with 50 mg/L bovine pituitary extract and 5 μg/L epidermal growth factor (EGF). All cell lines were grown in a humidified atmosphere of 5% CO_2_ at 37 °C.

### 4.3. Cell Proliferation Assay

Cells were seeded in 100 μL complete medium in a 96-well plate, at a density of 10,000 cells/well for LNCaP and 5000 cells/well for other cells. Cells were pre-incubated for 24 h to allow for adherence before removal of complete medium and treatment with various concentrations of compounds in fetal bovine serum (FBS)-free medium for 48 h. After treatment, 20 μL/well MTT (5 mg/mL) was added and incubated for 3–4 h at 37 °C, and then 100 μL/well DMSO was added for MTT assay. The plate was shaken on an orbital shaker for 10 min. Absorbance was measured using a microplate reader at 492 nm for the MTT assay (performed in triplicates). Cell viability was normalized to control. Half inhibitory concentration (IC_50_) was calculated by SPSS Statistics 17.0. The combination index (CI) isobologram equation was used for the quantitative determination of the interaction between two compounds [62]:(1)$$Combination\;index\;\left(\mathrm{CI}\right)\;=\frac{C_{A,x}}{IC_{x,A}}+\frac{C_{B,x}}{IC_{x,B}}$$
where *C_A,x_* and *C_B,x_* are compound concentrations that inhibit 50% cell growth compared to control when being used alone, and *IC_x,A_* and *IC_x,B_* are compound concentrations that provide the same effect when being used in combination. The experiments were independently repeated two to three times.

### 4.4. Transwell Assay

Cell migration was evaluated using 8-μm transwell filters (Costar Corning, Schiphol-Rijk, The Netherlands). Cells at a density of 5 × 10^4^ were seeded to the upper chamber, and the chamber was filled with FBS-free medium. Cells were incubated overnight to allow for adherence; then, the lower chamber was replaced with 10% FBS medium as a chemo-attractant. Following a 24 h incubation, non-migrating cells on the upper surface of the insert membrane were gently removed by cotton swabs and migrated cells on the lower surface of the insert membrane were fixed by 4% paraformaldehyde and stained with 0.1% crystal violet for 20 min. Migrated cells on each membrane were photographed and counted at 100× magnification field at six random fields.

### 4.5. Flow Cytometry

Cell apoptosis was assessed by Annexin V-FITC and propidium iodide (PI) staining using an Annexin V-FITC apoptosis detection kit (Thermo Fisher Scientific, Carlsbad, CA, USA) according to the manufacturer’s instructions. Briefly, cells were seeded at a density of 1.0 × 10^5^ cells/well and pre-incubated for 24 h to allow for adherence before treatment with various concentrations of tryptamine in an FBS-free medium. After 24 h of tryptamine incubation, cells in each well were harvested and centrifuged at 200× *g* for 5 min. Cells were washed with PBS and centrifuged at 200× *g* for 5 min. Cells were resuspended in 100 μL binding buffer, followed by staining with 5 μL Annexin V-FITC and 5 μL PI, and incubation on ice in the dark for 15 min after addition of each staining reagent. Subsequently, a 300 μL binding buffer was added to the sample and apoptotic cells were detected by a flow cytometer (NovoCyte; ACEA Biosciences Inc., San Diego, CA, USA).

### 4.6. Western Blot Assay

After treatment with various doses of tryptamine, PC-3 cells were lysed with RIPA lysis buffer (Beyotime, Shanghai, China) containing 1 mM PMSF, and the total protein concentration was determined by BCA Protein Assay Kit (Solarbio, Beijing, China). Samples were electrophoresed on a 10% sodium dodecylsulfate (SDS) polyacrylamide gel and electrotransferred onto polyvinylidene difluoride (PVDF) membrane (Beyotime, China). After blocking with 5% BSA for 1 h, the membrane was incubated with the primary antibodies rabbit anti-cleaved caspase-3 (1:1000, ab32042, Abcam) and anti-β-actin (1:1000, 4967S, Cell Signaling Technology, Boston, MA, USA) at 4 °C overnight, followed by incubation with horseradish peroxidase (HRP)-conjugated goat anti-rabbit secondary antibody (1:2000, 7074S, Cell Signaling Technology) for 1 h at 37 °C. After washing with TBST, the bands were visualized by the enhanced chemiluminescence kit (Thermo Fisher Scientific, Carlsbad, CA, USA) and Odyssey^®^ imaging system (LI-COR, Lincoln, NE, USA) according to the manufacturers’ instructions.

### 4.7. Confocal Imaging of Cells

PC-3 cells (5 × 10^4^ cells/well) were seeded in a 24-well chamber with a coverslip and incubated overnight. After treating with tryptamine (0.300 mM) for 8 h, the cells were washed with PBS and fixed with 4% paraformaldehyde at room temperature for 10 min, followed by 0.2% Triton X-100 for 5 min and blocked with 20% goat serum. The cells were incubated in rabbit anti-cleaved caspase-3 antibody (1:200, ab32042, Abcam) overnight at 4 °C, followed by incubation with secondary antibody Alexa Fluor 488-conjugated goat anti-rabbit IgG (1:500, A0423, Beyotime, Shanghai, China) for 1 h at room temperature in the dark. Cells were washed and stained with 4′,6-diamidino-2-phenylindole (DAPI). Fluorescence images were acquired with a laser scanning confocal microscope (LSCM, Leica, TCS SP8 MP, Heidelberg, Germany).

### 4.8. Xenograft Assays

Five-week-old male BALB/c nude mice were purchased from Charles River Laboratories (Beijing, China) and raised in a temperature and humidity-controlled environment (25 ± 2 °C, 50% ± 10% humidity) with a standard 12 h light-dark cycle with food and water in cages under germ-free conditions (SCXK2021-0006). All procedures involving animals and their care were approved by the Institutional Animal Care and Use Committee of the Ocean University of China in accordance with institutional and National Institutes of Health guidelines (OUC-SMP-2022-01-01). PC-3 cells (4 × 10^6^ dissolved in 150 μL PBS) were injected subcutaneously into the armpit area of each mouse. For the first treatment group, when tumors reached 50 mm^3^, 80 mg/kg of tryptamine dissolved in normal saline with 0.1% formic acid was administered intraperitoneally once every other day for 32 days (n = 6). For the second treatment group, when tumors reached 100 mm^3^, 1 mg/100 mm^3^ (dose calculated based on tumor volume) tryptamine was administrated intratumorally every day for 16 days (n = 5). The control groups were administrated with the vehicle. The tumor volume (mm^3^) was calculated based on the formula:(2)$$Tumor\;volume\;({\mathrm{mm}}^{3})=length\times width^{2}/2$$

Data are presented as mean ± standard deviation (SD). *p* values were calculated by Student’s *t*-test unless otherwise stated.

### 4.9. UPLC–HRMS-Based Metabolomics Study

#### 4.9.1. Sample Preparation for Cell Metabolomics

The cells were seeded in 10 cm Petri dishes at a density of 1 × 10^6^ cells/dish. After incubation overnight for adherence, cells were subjected to treatment with tryptamine (0.3 mM) in FBS-free medium, and control samples were treated with FBS-free medium. Metabolite extraction was performed when a change in either cell number or morphology was observed for the treated group compared to the control group. For cellular metabolite extraction, cells were first washed with PBS for three times, then 8 mL extraction solvent (methanol: acetonitrile: 0.5 M formic acid, 2:2:1 *v*/*v*/*v*) was added for quenching and cell lysis. The mixture was vortexed and sonicated on ice for 15 min to extract metabolites, and then centrifuged at 13,000 rpm for 10 min at 4 °C. The supernatant was collected and dried under vacuum. Before analysis, the sample was reconstituted in 100 μL H_2_O.

#### 4.9.2. Sample Preparation for Mouse Serum Metabolomics

Serum samples were collected from PC-3 xenograft mice that received intraperitoneal administration of tryptamine and the control group, respectively. In serum metabolite extraction, 300 μL of extraction solvent (acetone: acetonitrile: methanol, 1:1:1, *v/v/v*) was added to each serum sample (100 μL). Samples were vortexed for 20 s to aid in protein precipitation, and then centrifuged at 13,000 rpm for 5 min at 4 °C. After centrifugation, 800 μL of dichloromethane was added to 350 μL of supernatant and vortex-mixed. Following the addition of 250 μL of deionized water, samples were vortex-mixed again. The aqueous phase was used for metabolite analysis by UPLC–HRMS.

#### 4.9.3. UPLC–HRMS Instrument Settings

Metabolic profiling was performed using an Agilent 1290 Infinity UHPLC system fitted with a Waters ACQUITY UPLC BEH C_18_ column (2.1 × 50 mm, 1.7 μm particle size, Waters Corporation, Milford, CT, USA) coupled to an LTQ Orbitrap XL mass spectrometer (Thermo Fisher Scientific, Carlsbad, CA, USA). The chromatographic method involved elution with water with 0.1% acetic acid (A) and acetonitrile (B). The gradient program of cell metabolomics was as follows: 0–1 min 98–99% A, 1–3 min 2–30% A, 3–8 min 30–50% A, 8–10 min 50–95% A, 10–15 min 95–95% A. The flow rate was constant at 0.3 mL/min. The gradient program of serum metabolomics was as follows: 0–1 min 100–90% A, 1–2.5 min 90–85% A, 2.5–4 min 85–78% A, 4–6 min 78–62% A, 6–9 min 62–35% A, 9–12 min 35–20% A, 12–16 min 20–0% A, 16–18 min 0% A. The flow rate was constant at 0.25 mL/min from 0-12 min and changed to 0.3 at 16 min and 0.45 mL/min at 18 min. The column was maintained at 35 °C. After each run, the gradient was re-equilibrated to its initial conditions over 6 min. Sample injection volume was 10 μL. The main mass spectrometer parameters were set as follows: capillary temperature 300 °C; capillary voltage and tube lens 20 V and 35 V, respectively, for positive mode, and 40 V and 80 V, respectively, for negative mode; spray voltage 4.0 kV in positive mode, and 2.5 kV and 2.8 kV in negative mode for serum and cell metabolite profiling, respectively; auxiliary gas, sheath gas and sweep gas flow rate 10, 40, and 0 arbitrary units, respectively. Data were acquired in the *m/z* range of 50–1000. The resolving power was 60,000 at *m/z* 195. Ultra-high-purity argon (≥99.999%) was used as the collision gas in UPLC–MS/MS experiments, and normalized collision voltages were in the range of 15–55 V.

#### 4.9.4. Data Analyses

Raw data was pre-processed using Progenesis QI software version 2.4 (Nonlinear Dynamics, Waters Corporation, Milford, CT, USA), which included peak picking and alignment. A data matrix table containing metabolic peak identity (retention time (Rt and *m/z* pair)) and peak abundances in samples was obtained. We used our previously designed FORTRAN program for data post-processing, including analyses of adducts, in-source fragments and chloride isotopic ions, contaminant and artefact ion removal, QC-based scaling, and normalization. Principal components analysis (PCA) and oPLS-DA (orthogonal partial least squares-discriminant analysis) were applied for sample classification using the PLS Toolbox software (version 8.9.1, Eigenvector Research, Inc., Manson, WA, USA) based on Matlab R2021a (version 9.10.0, MathWorks, Natick, MA, USA). Differential metabolites were identified with *p*-value < 0.05 and variance importance in projection (VIP) value greater than 1. The metabolites were identified based on accurate mass matching to theoretical values within a 2 mDa tolerance and MS/MS matching with spectral data in Human Metabolome Database (HMDB) and/or manual analysis of MS/MS spectra (Appendix A). MetaboAnalyst 5.0 was used for metabolic pathway and network analyses.

## 5. Conclusions

Tryptophan metabolites including tryptamine were extensively studied for their roles in neurodegenerative disease and neurotransmission [12]; however, there is still a gap in the knowledge of their relationship with cancer and potential cytotoxic and antitumor effects. In this study, several tryptophan metabolites including tryptamine, 3-HAA, 3-HK, and CA were found to have inhibitory effects on PCa cell proliferation. Especially, tryptamine was demonstrated to have significantly higher cytotoxicity against PC-3 and LNCaP cells. For PC-3 xenograft mice, local intratumoral administration of tryptamine was shown to inhibit tumor growth and reduce the tumor size, while intraperitoneal administration of tryptamine promoted the tumor growth. The results showed that tryptamine might have a dual effect in regulating PCa tumor growth. 

The combined treatment of tryptamine + etoposide exhibited a synergistic inhibitory effect on PC-3 and LNCaP cell proliferation, while exerting antagonistic effects on RWPE-1 cells, indicating the potential use of tryptamine in adjunctive treatment of prostate cancer. Furthermore, tryptamine was demonstrated to induce cell apoptosis via the caspase-3-dependent pathway, and it also apparently inhibited the migration of PC-3, A549, HepG2, and PANC-1 cells. Cell metabolomics revealed several compounds with significant changes after tryptamine treatment, including decreased creatine, arginine, proline, aspartic acid, spermidine, spermine, and increased 5-HT.

The findings in this study expand our knowledge on the bioactivities of endogenous compounds in tumor growth and provide a strategy for discovering more endogenous compounds that could regulate tumor growth, filling in the empty spaces on the knowledge map of bioactive endogenous human metabolites, and helping us to better understand the roles of these compounds in tumor regulation, eventually contributing to ideas in prevention and treatment of cancer. Future work includes study of the detailed mechanism of cytotoxicity of tryptamine and the dual effect of tryptamine regulation on PC-3 tumor growth with respect to intratumoral and intraperitoneal treatments to assess its potential use in treatment of PCa tumors.

## Figures and Tables

**Figure 1 ijms-23-11087-f001:**
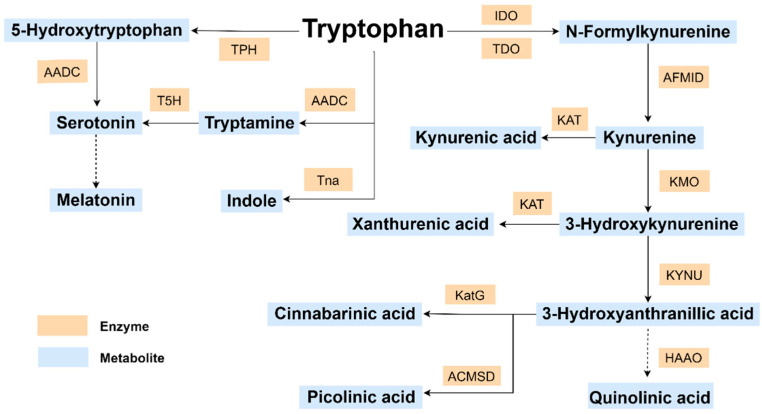
Tryptophan metabolism pathway. Key enzymes: IDO: indoleamine 2,3-dioxygenase; TDO: tryptophan 2,3-dioxygenase; AFMID: arylformamidase; KAT: kynurenine aminotransferase; KYNU: kynureninase; KatG: catalase-peroxidase; KMO: kynurenine 3-monooxygenase; ACMSD: aminocarboxymuconate-semialdehyde decarboxylase; HAAO: 3-hydroxyanthranilate 3,4-dioxygenase; TPH: tryptophan 5-monooxygenase; T5H: tryptamine 5-hydroxylase; Tna: tryptophanase; AADC: aromatic L-amino acid decarboxylase.

**Figure 2 ijms-23-11087-f002:**
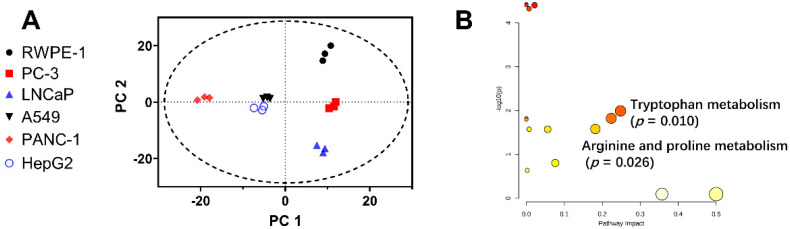
Alteration of tryptophan metabolism pathway between PCa and normal prostate cell lines. (**A**) PCA analysis of UPLC–HRMS metabolic profiles of PC-3, LNCaP, RWPE-1, A459, HepG2, and PANC-1 cells. (**B**) Pathway analysis revealed that tryptophan metabolism was significantly altered between PCa cells (PC-3 and LNCaP) and RWPE-1 cells.

**Figure 3 ijms-23-11087-f003:**
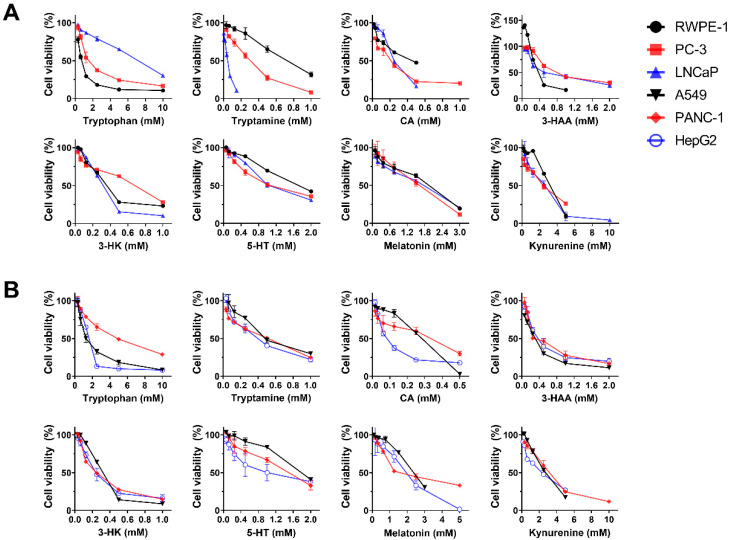
Cytotoxicities of major metabolites of the tryptophan pathway. MTT cell proliferation assays of various concentrations of tryptophan, tryptamine, CA, 3-HAA, 3-HK, 5-HT, melatonin, and kynurenine in (**A**) PC-3, LNCaP, and RWPE-1, and (**B**) A549, PANC-1 and HepG2 cells.

**Figure 4 ijms-23-11087-f004:**
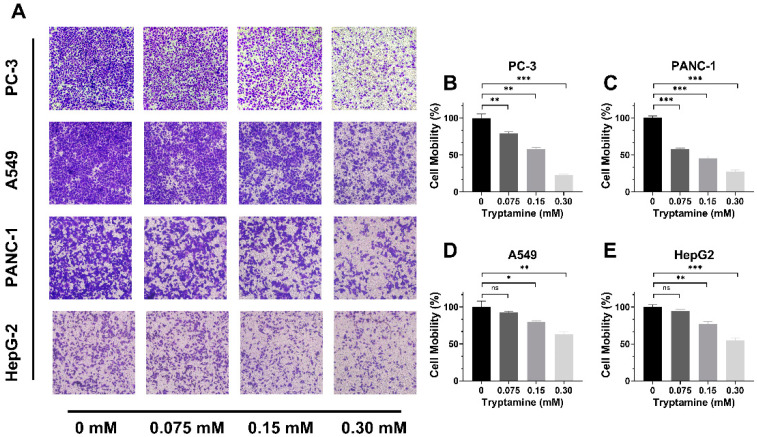
Migration of PC-3, PANC-1, A549, and HepG2 cells after treatment with different concentrations of tryptamine for 24 h. (**A**) Migrated cancer cells photographed at 100× magnification field. (**B**–**E**) Mobilities of PC-3, PANC-1, A549, and HepG-2 cells after incubation of different concentrations of tryptamine for 24 h. ns *p* > 0.05, * *p* < 0.05, ** *p* < 0.01, *** *p* < 0.001, n = 3.

**Table 1 ijms-23-11087-t001:** Cytotoxicities of studied endogenous metabolites in six cell lines.

Compound	IC_50_ Value (Mean ± SD mM)					
	RWPE-1	PC-3	LNCaP	A549	HepG2	PANC-1
Tryptophan	0.76 ± 0.03	1.89 ± 0.13	6.51 ± 0.29	1.50 ± 0.15	1.54 ± 0.04	4.54 ± 0.19
Tryptamine	0.71 ± 0.06	0.24 ± 0.01	0.037 ± 0.001	0.46 ± 0.01	0.35 ± 0.02	0.38 ± 0.01
3-HAA	0.38 ± 0.01	0.90 ± 0.04	0.65 ± 0.06	0.27 ± 0.01	0.42 ± 0.02	0.41 ± 0.01
Melatonin	1.47 ± 0.10	1.28 ± 0.06	1.25 ± 0.12	2.31 ± 0.06	1.48 ± 0.10	2.02 ± 0.09
Kynurenine	2.79 ± 0.05	2.02 ± 0.05	1.78 ± 0.18	2.46 ± 0.05	1.93 ± 0.03	2.63 ± 0.04
5-HT	1.66 ± 0.01	1.07 ± 0.03	1.12 ± 0.04	1.89 ± 0.09	1.05 ± 0.17	1.34 ± 0.13
3-HK	0.36 ± 0.01	0.55 ± 0.01	0.29 ± 0.01	0.29 ± 0.01	0.25 ± 0.01	0.25 ± 0.01
CA	0.40 ± 0.01	0.17 ± 0.01	0.24 ± 0.01	0.21 ± 0.01	0.10 ± 0.01	0.25 ± 0.04
5-HTP	0.85 ± 0.05	0.99 ± 0.04	1.80 ± 0.17	-	-	-
KYNA	4.39 ± 0.10	8.76 ± 1.18	9.74 ± 1.30	-	-	-
QA	3.20 ± 0.16	6.00 ± 1.32	6.83 ± 0.42	-	-	-
Picolinic acid	1.48 ± 0.06	4.95 ± 0.19	5.75 ± 034	-	-	-
Indole	0.90 ± 0.06	0.70 ± 0.02	0.98 ± 0.02	-	-	-
Tyramine	2.99 ± 0.08	1.53 ± 0.04	1.71 ± 0.12	3.18 ± 0.24	1.53 ± 0.05	0.77 ± 0.04
β-phenylethylamine	2.51 ± 0.19	0.44 ± 0.01	0.69 ± 0.06	0.72 ± 0.06	1.01 ± 0.03	1.47 ± 0.07
Etoposide	0.016 ± 0.001	0.064 ± 0.001	0.032 ± 0.001	0.041 ± 0.002	0.030 ± 0.001	0.042 ± 0.001

**Table 2 ijms-23-11087-t002:** CI values of the combined treatments.

Compound Combination	CI Value					
	RWPE-1	PC-3	LNCaP	A549	HepG2	PANC-1
Tryptamine + 5-HT	1.38	0.91	1.06	0.90	0.86	0.98
Tryptamine + 3-HAA	1.98	1.04	0.92	1.20	0.47	0.74
Etoposide + Tryptamine	1.25	0.78	0.78	0.98	1.29	0.84
Etoposide + 5-HT	1.30	0.70	0.55	0.89	1.37	0.94
Etoposide + 3-HAA	3.03	0.80	1.05	0.93	0.82	0.51

**Table 3 ijms-23-11087-t003:** Tentative identification of differential metabolites of PC-3 cells between control and tryptamine treatment groups.

Compound Name	Retention Time (min)	*m*/*z*	Elemental Formula	Δm (mDa)	Ion Type	MS/MS ^a^	*p*	Fold Change ^b^
arginine	0.51	175.1189	C_6_H_14_N_4_O_2_	0.7	[M+H]^+^	158.0922/130.0972	0.035	−1.9
proline	0.64	116.0700	C_5_H_9_NO_2_	1.1	[M+H]^+^	98.0593/70.0644	0.033	−2.0
creatine	0.64	132.0763	C_4_H_9_N_3_O_2_	0.4	[M+H]^+^	90.0543	2.5 × 10^−4^	−2.0
spermidine	0.49	146.1649	C_7_H_19_N_3_	0.8	[M+H]^+^	129.1382/112.1115/72.0801	5.7 × 10^−4^	−2.1
spermine	0.49	203.2230	C_10_H_26_N_4_	0.6	[M+H]^+^	129.1382	4.2 × 10^−5^	−2.1
5-HT	2.68	177.1018	C_10_H_12_N_2_O	0.3	[M+H]^+^	160.0753	1.4 × 10^−3^	1.4 × 10^3^
pantothenic acid	3.61	218.1032	C_9_H_17_NO_5_	0.4	[M-H]^−^	88.0402	0.045	−1.6
aspartic acid	0.62	132.0302	C_4_H_7_NO_4_	0.1	[M-H]^−^	115.0031/88.0402	2.2 × 10^−3^	−3.2
α-ketoisovaleric acid	0.14	115.0395	C_5_H_8_O_3_	0	[M-H]^−^	71.0503	0.040	0
ketoleucine	2.37	129.0549	C_6_H_10_O_3_	0.1	[M-H]^−^	111.0446/83.0500	3.1 × 10^−3^	0

^a^ MS/MS spectra of all features were matched with the HMDB MS/MS spectra and/or manually predicted MS/MS fragments within 5 mDa; *p*-value is calculated by Student’s *t*-test; ^b^ fold change is the average tryptamine treatment/control ratio, with positive and negative values indicating increased and decreased levels in the treatment group, respectively.

## Data Availability

All datasets are shared in FigShare at https://doi.org/10.6084/m9.figshare.20646642.v1.

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
