# Peer review of "Dual Effect of Tryptamine on Prostate Cancer Cell Growth Regulation: A Pilot Study"

_ijms, 2022, doi:10.3390/ijms231911087_

Round 1

Reviewer 1 Report

 The manuscript submitted by Li et al,.  is interesting, well designed and well written, the following comments could improve the manuscript:

1. The introduction is too lengthy and should be shortened, and many information included in the introduction could be included in the discussion section.

2. The concentration used should be unified in the result section (either using mM or µM and not alternating among them).

3. The number of mice in each group should be included in the materials and methods

Author Response

Response to Reviewer 1 Comments

Point 1: The introduction is too lengthy and should be shortened, and many information included in the introduction could be included in the discussion section.

Response 1: We thank the reviewer for his/her suggestion. We shortened the introduction section and adjusted some of the content to the discussion section.

Point 2: The concentration used should be unified in the result section (either using mM or µM and not alternating among them).

Response 2: Thank the reviewer for pointing out this. We unified the concentration with “mM” in the revised manuscript.

Point 3: The number of mice in each group should be included in the materials and methods.

Response 3: We thank the reviewer for the rigorous evaluation regarding to the terminology in our study. We added the number of mice in 4.8 Xenograft assays section:

    “For the first treatment group, when tumors reached 50 mm3, 80 mg/kg of tryptamine dissolved in normal saline with 0.1% formic acid was administered intraperitoneally once every other day for 32 days (n=6). For the second treatment group, when tumors reached 100 mm3, 1 mg/100 mm3 (dose calculated based on tumor volume) tryptamine was administrated intratumorally every day for 16 days (n=5).”

Reviewer 2 Report

Dear authors,

Manuscript ijms-1910517 entitled "Screening for bioactive tryptophan metabolites affecting prostate cancer cell growth - tryptamine as a potential candidate" and authored by Zhuangzhuang Li, Baoyan Ding, Mustafa R. K. Ali, Lizhen Zhao, Xiaoling Zang and Zhihua Lv targets a hot topic that is potentially very interesting to journal readers and to the scientific community as a whole. The study have been accurately designed and nicely conducted. The results improve our understanding of tryptophan bioactivity on prostate cancer development. while I am very supportive for the publication of the findings I feel that the paper in its current form could not be accepted I rather suggest a revision that addresses the follwing points that needs to be aimproved:

1. The title : the title is not very attractive to readers I find it a passive title that not encourage me personally to read the manuscript. I rather suggest a more informative title that reports the findings.

2. It is impossible for me to recommad the paper for publication with the low resolution of figures. It is pitty that I can not read figure 3B. All figures have to be checked and the resolution improved.

3. The conclusion is really very passive. Why the authors do not highlight their fiondings in a more proactive way. Why the impacts of the study are not nicely highlighted. Authors should identify future directions and experiments that needs to be conducted in the near future. Why gaps in our knowledge are not discussed.

I will be happy to read a new version of the manuscript that addresses all these drawbacks and that i canm recommand for publication.

Best regards

Author Response

Response to Reviewer 2 Comments

Point 1: The title: the title is not very attractive to readers I find it a passive title that not encourage me personally to read the manuscript. I rather suggest a more informative title that reports the findings.

 Response 1: We appreciate the reviewer’s insightful comment. In the revised manuscript, we used “Dual effect of tryptamine on prostate cancer cell growth regula-tion - a pilot study” instead of the previous title.

Point 2: It is impossible for me to recommad the paper for publication with the low resolution of figures. It is pitty that I can not read figure 3B. All figures have to be checked and the resolution improved.

Response 2: We are very sorry for our negligence of the figure resolution. We rearranged and provided high-resolution figures with 600 dpi.

Point 3: The conclusion is really very passive. Why the authors do not highlight their fiondings in a more proactive way. Why the impacts of the study are not nicely highlighted. Authors should identify future directions and experiments that needs to be conducted in the near future. Why gaps in our knowledge are not discussed.

Response 3: We thank the reviewer for his/her suggestion. In the conclusion section, we added more discussion as follows

“Tryptophan metabolites including tryptamine were extensively studied for their roles in neurodegenerative disease and neurotransmission [1], however, there is still a gap in the knowledge of their relationship with cancer and potential cytotoxic and antitumor effects. In this study, several tryptophan metabolites including tryptamine, 3-HAA, 3-HK and CA were found to have inhibitory effects on PCa cell proliferation. Especially, tryptamine was demonstrated to have significantly higher cytotoxicity against PC-3 and LNCaP cells. For PC-3 xenograft mice, local in-tratumoral administration of tryptamine was shown to inhibit tumor growth and reduce tumor size, while intraperitoneal administration of tryptamine promoted the tumor growth. The results showed that tryptamine might have a dual effect in regulating PCa tumor growth.” (on page 16).

[1] Platten, M.; Nollen, E.A.; Röhrig, U.F.; Fallarino, F.; Opitz, C.A. Tryptophan metabolism as a common therapeutic target in cancer, neurodegeneration and beyond. Nat. Rev. Drug Discovery 2019, 18, 379-401.

Round 2

Reviewer 2 Report

Dear authors,

The new version of the manuscript is satisfactory. I recommend publication of the manuscript

Best regards